# Direct Pattern Growth of Carbon Nanomaterials by Laser Scribing on Spin-Coated Cu-PI Composite Films and Their Gas Sensor Application

**DOI:** 10.3390/ma14123388

**Published:** 2021-06-18

**Authors:** Yong-il Ko, Geonhee Lee, Min Jae Kim, Dong Yun Lee, Jungtae Nam, A-Rang Jang, Jeong-O Lee, Keun Soo Kim

**Affiliations:** 1Department of Physics and Graphene Research Institute, Sejong University, Seoul 05006, Korea; armist130@gmail.com (Y.-i.K.); geonhl@sejong.ac.kr (G.L.); mjkimphys@gmail.com (M.J.K.); geovi012@gmail.com (D.Y.L.); goodnjt@kist.re.kr (J.N.); 2Department of Electrical Engineering, Semyung University, Jecheon 27136, Korea; 3Advanced Materials Division, Korea Research Institute of Chemical Technology (KRICT), Gajeong-ro 141, Daejeon 34114, Korea; jolee@krict.re.kr

**Keywords:** carbon nanomaterials, laser scribing, polyimide, copper particle, gas sensor

## Abstract

The excellent physical and chemical properties of carbon nanomaterials render them suitable for application in gas sensors. However, the synthesis of carbon nanomaterials using high-temperature furnaces is time consuming and expensive. In this study, we synthesize a carbon nanomaterial using local laser-scribing on a substrate coated with a Cu-embedded polyimide (PI) thin film to reduce the processing time and cost. Spin coating using a Cu-embedded PI solution is performed to deposit a Cu-embedded PI thin film (Cu@PI) on a quartz substrate, followed by the application of a pulsed laser for carbonization. In contrast to a pristine PI solution-based PI thin film, the laser absorption of the Cu-embedded PI thin film based on Cu@PI improved. The laser-scribed carbon nanomaterial synthesized using Cu@PI exhibits a three-dimensional structure that facilitates gas molecule absorption, and when it is exposed to NO_2_ and NH_3_, its electrical resistance changes by −0.79% and +0.33%, respectively.

## 1. Introduction

Carbon nanomaterials are core materials for a wide range of industries owing to their excellent physical and chemical properties. Their molecular adsorption behavior resulting from their high specific surface areas and conductivity enables their application in gas sensors [1,2,3]. Their synthesis typically involves heat treatment at ~1000 °C or higher, which involves lengthy processes and undesired costs [4,5,6]. By contrast, the laser-scribing method enables the synthesis of carbon nanomaterials in a shorter processing time (s) and carbonization in a desired pattern; hence, its application has been extensively investigated [7,8,9,10]. Polyimide (PI) is a crystalline polymer with excellent resistance to heat, chemicals, and abrasion. PI comprises an imide ring, which facilitates pyrolysis, and has been extensively used in high-quality carbon materials [11,12,13]. However, previous studies regarding laser-scribed carbon nanomaterials indicated limitations in their applications owing to the use of polymer substrates, including issues pertaining to durability and device fabrication. Hence, the extensive application of laser-scribed carbon nanomaterials requires a transfer process to the target substrate [14].

In this study, a Cu-embedded PI thin film was prepared on a quartz substrate via spin coating using a Cu-embedded PI solution. Because the spin-coated PI thin film has minimal absorbance and transmits most of the laser beam, Cu particles, which improve laser absorption and serve as a catalyst in pyrolysis, were complexed with PI (Cu@PI), and a Cu–laser-scribed carbon composite (Cu@LSC) was successfully synthesized using laser scribing in a vacuum chamber [15,16,17]. Hence, a carbon nanomaterial channel was patterned for gas sensing in the desired position and shape without requiring a transfer process. The synthesized Cu@LSC exhibited a higher sensitivity for NO_2_ than for NH_3_, since Cu particles serve as electron donors [18], which accelerate the electron-donating effect of the sensor. Because quartz is non-conducting, transparent, and resistant to heat, it can be used as a stable supporting substrate for laser scribing [8,19,20] as well as a gas sensor [21,22]. Therefore, Cu@LSC on a quartz substrate should operate as a stable gas sensor under extreme conditions, such as fires and other disasters. The chemical and structural properties of the prepared materials were evaluated using Raman spectroscopy, X-ray photoelectron spectroscopy (XPS), optical microscopy (OM), and scanning electron microscopy (SEM). The gas-sensing performance was measured to evaluate the dynamic response to NO_2_ and NH_3_ gases using a source meter.

## 2. Materials and Methods

### 2.1. Preparation of Cu@PI Thin Film Using Cu-Embedded PI Solution

A schematic illustration for the preparation of the Cu@LSC is shown in Figure 1. Cu powder (<100 nm, Sigma–Aldrich, St. Louis, MO, USA) was added to N-methyl-2-pyrrolidone (NMP, Samchun Chemicals, Seoul, Korea) at 20 wt.% and sonicated for 30 min. Solutions of Cu-NMP and PI (Vtec PI solution 1388,20 wt.% in NMP, Standard Systems, Seoul, Korea) were mixed at weight ratios of 2.5, 5.0, 7.5 and 10.0 wt.%, and then stirred for 30 min. Next, 300 μL of the Cu@PI mixture was deposited onto a clean quartz plate (15 mm × 15 mm × 1 mm, Hanjin Quartz, Seoul, Korea) and spin coated at 1000–2500 rpm for 30 s. Evaporation was performed sequentially at 100 °C (2 min), 120 °C (1 min), and 150 °C (5 min), and curing was performed at 325 °C (15 min) after treatment at 200 °C (15 min).

### 2.2. Laser-Scribing on Cu@PI Thin Film and Peeling off Unexposed Area

The Cu@PI thin film on a quartz plate was laser scribed using a pulsed ytterbium fiber laser (Maxphotonics Co., Ltd., MFP-20, Shenzhen, China). The specifications are detailed in the Supporting Information (Appendix A and Appendix A). The scan rate, spot distance, laser power, and repeat count were controlled to prepare the optimized Cu@LSC under vacuum. The Cu@LSC was immersed in ammonia (20–30%, Duksan Chemicals, Ansan, Korea) for 1 h to remove unexposed Cu@PI; consequently, only the carbonized region remained and was used to create a carbon nanomaterial channel (1 mm × 5 mm) for gas detection.

### 2.3. Gas Sensing

A Cu@LSC gas sensor was used to evaluate the sensitivity to NO_2_ and NH_3_ gases using a source meter (Keithley 236, Cleveland, OH, USA) and a switch system (Keithley 708A) (Appendix A). A 0.1 V direct current (DC) potential was applied to the sensor, and changes in its resistance level were continuously monitored at room temperature and atmospheric pressure. The Cu@LSC-based chemical sensor experiments were performed based on the selected concentrations of the target gases diluted with air at a total gas flow of 500 sccm. The injection and concentration of the gases were controlled automatically using a mass flow controller, and the durations of the target and background gases were set to 10 and 50 min, respectively. The normalized sensor response is defined as follows:ΔR/R_0_ (%) = (R − R_0_)/R_0_ × 100%(1)
where R_0_ and R are the resistances of the sensor in air and the target gas, respectively.

### 2.4. Characterization

Visible–near-infrared (Vis–NIR) spectra were recorded using an ultraviolet–visible–near-infrared (UV–Vis–NIR) spectrometer (UV-1800PC, AOE Instruments, Shanghai, China). Raman analysis, including OM, was performed using an in-Via Raman spectroscope with a 514 nm excitation line (Renishaw, Gloucestershire, UK). Chemical and morphological analyses were performed using SEM (TESCAN VEGA3, Brno–Kohoutovice, Czech Republic) and XPS (K-alpha (Al Kα), Thermo Fisher, Waltham, MA, USA).

## 3. Results and Discussion

### 3.1. Effect of Cu Particles

As shown in the Vis–NIR spectra in Figure 2a, the PI (without Cu particles) and Cu@PI (10 wt.% of Cu) films demonstrated transmittances of 95.9% and 21.9%, respectively, at a laser wavelength (rotational speed = 1000 rpm). The photographs of the laser scribed area (Figure 2a inset) from the PI film showed carbon deficiency; however, Cu@PI clearly showed carbonization (in black) because of the increased absorption of the laser owing to the Cu particles. Hence, the local area exposed to the laser reached carbonization temperatures, and Cu served as a catalyst for pyrolysis [15,16]. This is visible in the Raman spectrum of Cu@LSC (Figure 2b), which clearly shows first-order scattered G, D, and second-order scattered (2D) bands attributed to carbon [23,24], whereas weak bands were observed in the corresponding LSC from the PI film.

To verify the effect of the Cu particles more comprehensively, we conducted an experiment by changing the ratio of Cu. Consequently, the transmittance decreased from 75.3 to 21.9 as the weight ratio of Cu increased from 2.5 to 10.0 wt.% (Figure 3a). This implies that the absorption of the laser increased at a higher ratio of Cu, and that carbonization can be performed at a higher energy level. Figure 3b and Appendix A show the Raman spectra and calculated I_D_/I_G_ and I_2D_/I_G_ of Cu@LSC with different weight ratios (rotational speed = 1000 rpm; scan rate = 450 mm/s; laser power = 4 W). In general, in the Raman analysis of carbon materials, the D and 2D bands indicate disorder-induced first-order scattering and second-order scattering of the graphitic structure, respectively. Hence, the carbonization state can be compared by investigating the changes in I_D_/I_G_ and I_2D_/I_G_ [23,24,25]. When the weight ratio was 2.5 wt.%, I_D_/I_G_ was 1.57, and the 2D band was almost invisible. This implies that it comprises an almost amorphous structure containing significant disorders. By contrast, the Raman spectrum of Cu@LSC at 10.0 wt.% exhibited sharpened G, D, and 2D bands, and I_D_/I_G_ decreased to 0.52, which was lower than those of 5.0 wt.% (0.95) and 7.5 wt.% (0.57). In addition, I_2D_/I_G_ was 0.16, 0.69, and 0.71, at 5.0, 7.5, and 10.0 wt.%, respectively. Therefore, high-quality carbon was synthesized at a higher ratio of Cu, and it was confirmed that Cu particles served as a catalyst to form an enhanced graphitic structure with less disorder. However, the spin-coating quality was low at Cu ratios exceeding 10 wt.% because of the rapidly increasing viscosity. Therefore, the optimal concentration was achieved when the ratio of Cu was 10 wt.%.

### 3.2. Spin-Coating Conditions of Cu@PI Thin Film

Figure 4a shows the Vis–NIR spectrum when the rotational speed of Cu@PI on a quartz substrate was changed from 1000 to 2000 rpm (weight ratio of Cu = 10 wt.%; scan rate = 450 mm/s; laser power = 4 W). The transmittance was 21.9%, 40.3%, 63.5%, and 81.4% at rotational speeds of 1000, 1500, 2000, and 2500 rpm, respectively. This change in the laser absorption affected the carbonization quality. The relevant Raman spectra and calculated I_D_/I_G_ with respect to the change in the rotational speed are shown in Figure 4b and Appendix A. The I_D_/I_G_ of the thinnest film at 2500 rpm (thickness: 13 μm) was 1.05, indicating that the defect-induced D band was large, and the 2D band representing the graphitic structure was small. By contrast, at 1000 rpm (thickness: 44 μm), the I_D_/I_G_ ratio decreased to 0.52, and both the G and 2D bands increased, indicating that a decrease in the rotational speed enhanced carbonization via an increase in laser absorption, and that the Cu particles served as a catalyst [15,17].

### 3.3. Optimization of Laser-Scribing Process (Scan Rate and Laser Power)

The distance of the laser spot was calculated using the scan rate and frequency, as shown in Equation (2). At a fixed frequency of 30 kHz and adjusted scan rates of 270, 450, and 660 mm/s, the spot spacings were 9, 15, and 22 μm, respectively. At a low rate of 270 mm/s, the spot spacing reduced, thereby enabling the overlapping region of the laser irradiation to increase (Appendix A). By contrast, at a high rate of 660 mm/s, the spot spacing increased, and a non-scribed portion appeared. Therefore, the laser scan rate and spacing conditions must be optimized:Distance (μm) = scan rate (mm/s)/frequency (kHz)(2)

To verify the effect of the scan rate on the Cu@PI surface, we evaluated Cu@LSC before peel-off using OM and Raman analysis. The OM image obtained at a speed of 270 mm/s (Appendix A) shows that the quartz substrate was exposed because of excessive combustion due to energy accumulation; the Raman spectra (Figure 5a) support this finding. Weak D and G bands were observed at 270 mm/s along with a quartz baseline with significant noise. At 450 mm/s, the scribed part is carbonized (black area shown in Appendix A), whereas clear D, G, and 2D bands are visible in the Raman spectra (shown in Figure 5a), confirming graphite-type carbonization [25]. However, the surface shape of the PI film remained unchanged when the scan rate was increased to 660 mm/s (Appendix A), and a significantly weak carbon-related band was observed (Figure 5a) because the spots could not be connected, and incomplete carbonization occurred.

Figure 5b shows the Raman spectra of Cu@LSC after peel-off. When the scan rates were 270 and 450 mm/s, a signal similar to that observed before peel-off (Figure 5a) was observed. By contrast, at 660 mm/s, only a quartz signal was observed because Cu@LSC could not settle on the surface of the quartz substrate owing to insufficient carbonization in the axial direction. Therefore, the scan rate of 450 mm/s was optimal for patterning and maintaining the shape stability.

Figure 6 shows the Raman spectra based on the laser power (experimental conditions; I_D_/I_G_ and I_2D_/I_G_ are listed in Appendix A). When the repeat count was fixed at three and the laser power was 2 and 3 W, distinct Raman signals could not be observed, indicating that the Cu@PI film was no longer deeply carbonized in the vertical direction. This phenomenon occurred because Cu@LSC could not settle on the quartz surface, and the carbonized section peeled off along with the untreated section (Appendix A). By contrast, when the laser power was 4 W, the Raman spectra of Cu@LSC exhibited sharpened G, D, and 2D bands, and I_D_/I_G_ and I_2D_/I_G_ were 0.52, and 0.71, respectively (Figure 6). As shown in Appendix A, the laser-treated part of Cu@PI was completely carbonized in the vertical direction and settled onto the quartz surface, thereby maintaining its shape even after immersion in ammonia. However, the carbon-related bands broadened in the case of 5 W; hence, I_D_/I_G_ increased to 0.85, and the 2D band almost disappeared. These results indicate that the graphitic structure was decomposed and converted into an amorphous structure because of excessive laser exposure. Therefore, a laser power of 4 W is optimal for patterning, maintaining shape stability, and guaranteeing high quality.

### 3.4. Morphological and Chemical Analysis of Cu@LSC

The SEM images (Figure 7) show the morphology of Cu@LSC under optimized scribing conditions (weight ratio of Cu = 10 wt.%; spin coating = 1000 rpm; scan rate = 450 mm/s; laser power = 4 W). A low-magnification image of the patterned carbon nanomaterial on the quartz substrate is shown in Figure 7a. Cu@LSC with a diameter of 500 μm and the quartz substrate after peeling off the unexposed Cu@PI region were observed. The Cu@LSC tilted at 70° under high magnification (Figure 7b) exhibited a stereoscopic three-dimensional (3D) structure as an aggregate of carbon nanomaterials rather than a flat one. Therefore, this structure is desirable for gas-sensing applications because it can easily capture and adsorb target gases [26,27,28].

Figure 8 shows the XPS spectra of Cu@PI and Cu@LSC. The C1s spectrum of Cu@PI exhibited a maximum intensity at 285.6 eV because a large portion of the sp3 C–C bond and the C=O peak were observed at 288.8 eV. The C1s spectrum of Cu@LSC shifted to 285.2 eV owing to the significantly increased sp2 C=C portion, and the C=O peak almost disappeared. The atomic% of C1s increased from 79.34% to 83.51%, indicating that carbonization with a graphitic structure proceeded through laser scribing. In addition, Cu2p1/2 and Cu2p3/2 peaks were observed before and after laser scribing, respectively, and the atomic% of Cu2p increased from 0.63% to 0.83%, proving that the Cu particles did not decompose during laser scribing and served as a catalyst.

### 3.5. Gas Sensing Performance

Figure 9 shows the NO_2_ and NH_3_ response curves of the directly synthesized Cu@LSC on a quartz substrate at room temperature and atmospheric pressure. Figure 10 shows the resistance changing rates and resistance response vs. time. Generally, NO_2_ molecules withdraw free electrons from oxygen-terminated groups such as carboxyl and hydroxyl groups, and NH_3_ gas molecules donate electrons to the protonated groups on the sensor surface. Hence, the resistance of the sensor changes in opposite directions, i.e., decreasing with NO_2_ and increasing with NH_3_ [29]. In the case of NO_2_ gas flow for 10 min, the resistance responses detected were −0.29, −0.53, and −0.79% during exposure to 10, 50, and 100 ppm of NO_2_ gas, respectively. This is attributable to a carbonized structure having a suitable oxygen derived defect that causes sensing of target gas, as shown in Raman and XPS results. And the stereoscopic 3D structure having a large surface area (Figure 7) facilitates the adsorption/desorption of gas molecules on the surface [26,27,28]. Furthermore, because Cu nanoparticles facilitate the chemical adsorption of NO_2_ on the surface of carbon by supplying free electrons [18], Cu is expected to improve the sensor performance of Cu@LSC. In the case of NH_3_ gas, the sensing response was +0.13, +0.19, and +0.33% with exposure to 10, 50, and 100 ppm of NH_3_ gas, respectively. Therefore, the sensitivity of the Cu@LSC-based sensor was higher to NO_2_ than to NH_3_.

## 4. Conclusions

In this study, we fabricated Cu@LSC with a desired shape and position from the carbonization of a Cu@PI film via spin coating and laser scribing, which is cheaper and faster than conventional methods. In particular, the absorption during laser scribing and the sensitivity of the target gas were enhanced by the addition of Cu nanoparticles. The weight ratio of Cu, optimal rotational speed for spin coating, scan rate, and laser power during laser scribing were 10 wt.%, 1000 rpm, 450 mm/s, and 4 W, respectively. Raman results indicated that the I_D_/I_G_ and I_2D_/I_G_ values of the synthesized Cu@LSC were 0.52 and 0.71, respectively, and 83.51 at.% of C1s was observed via XPS. Furthermore, the synthesized Cu@LSC exhibited a 3D structure that facilitated the adsorption of gas molecules. When the fabricated device was exposed to NO_2_ (NH_3_), changes in electrical resistance by −0.29 (+0.13)%, −0.53 (+0.19)%, and −0.79 (+0.33)% were observed during exposure to 10, 50, and 100 ppm of NO_2_ (NH_3_), respectively. The Cu@LSC-based sensor was thermally stable because the carbon materials were patterned on quartz; hence, it is expected to function as an outstanding gas sensor even under extreme environmental hazard conditions.

## Figures and Tables

**Figure 1 materials-14-03388-f001:**
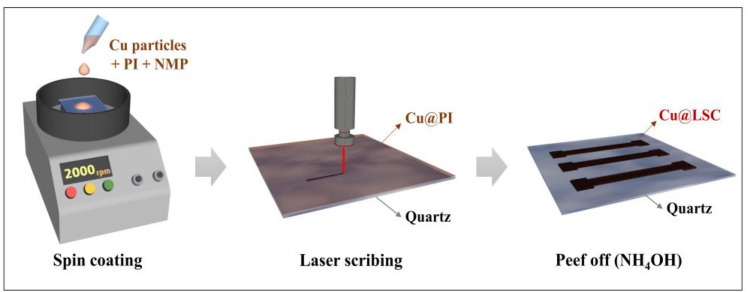
Schematic illustration of synthesis process of Cu@LSC.

**Figure 2 materials-14-03388-f002:**
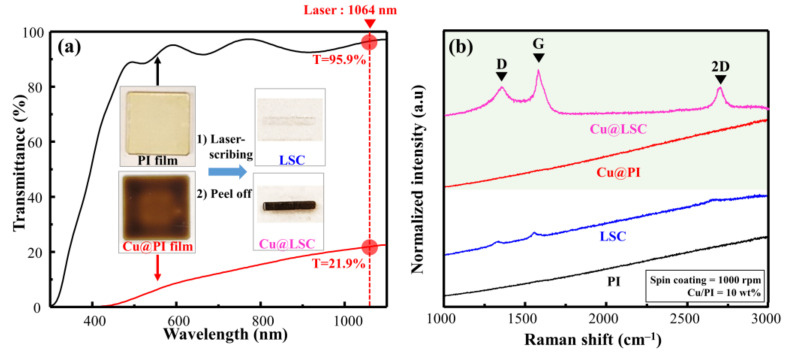
(**a**) VisNIR spectra and photographs, and (**b**) Raman spectra of spin-coated raw PI and Cu@PI film with laser scribing.

**Figure 3 materials-14-03388-f003:**
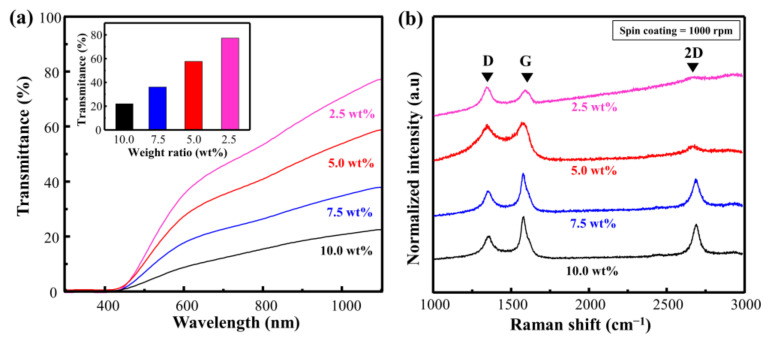
(**a**) Vis–NIR spectra of spin-coated Cu@PI film and (**b**) Raman spectra of Cu@LSC with different condition of weight ratios of Cu to PI.

**Figure 4 materials-14-03388-f004:**
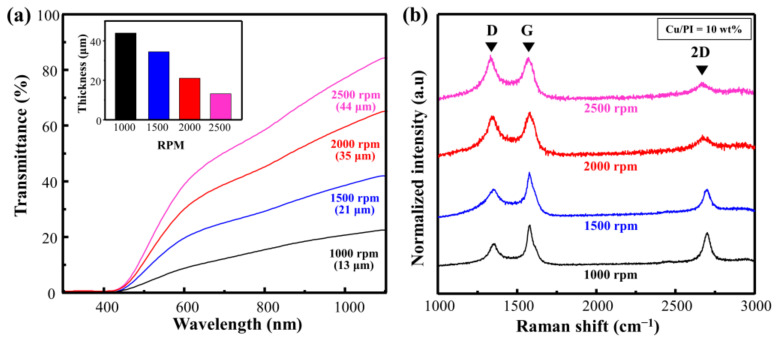
(**a**) Vis–NIR spectra of spin-coated Cu@PI film and (**b**) Raman spectra of Cu@LSC with different spin-coating conditions.

**Figure 5 materials-14-03388-f005:**
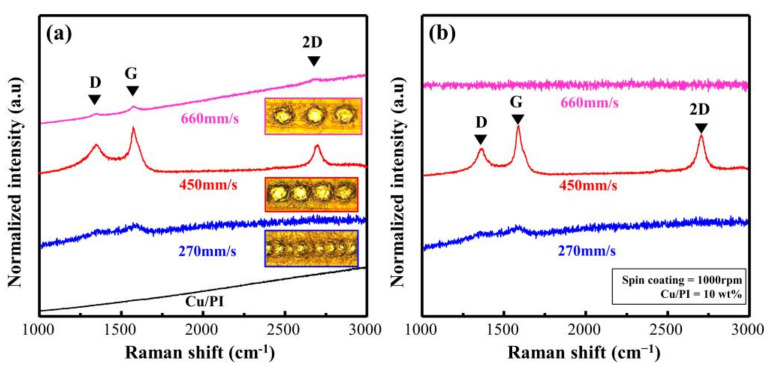
Raman spectra of Cu@LSC with different scan rates (**a**) before and (**b**) after peel-off.

**Figure 6 materials-14-03388-f006:**
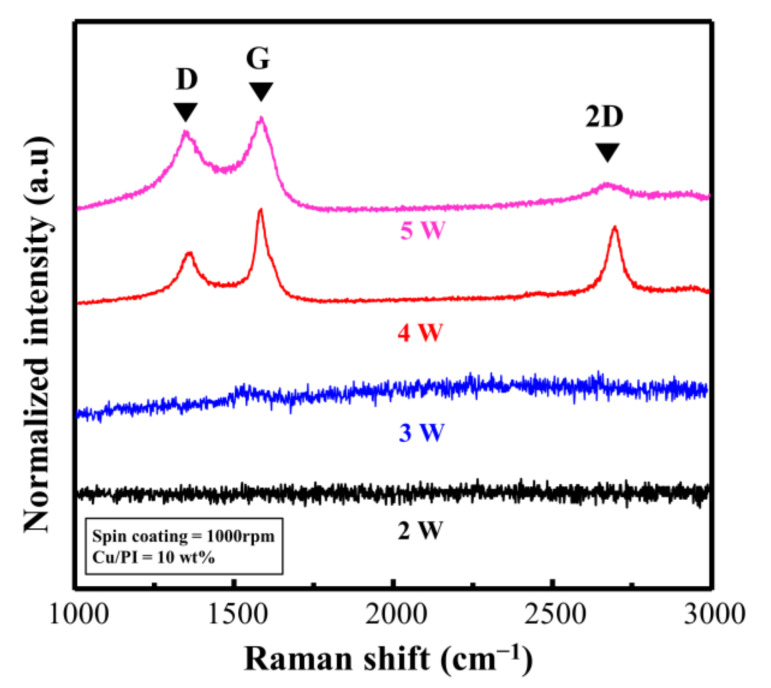
Raman spectra of Cu@LSC with different laser powers.

**Figure 7 materials-14-03388-f007:**
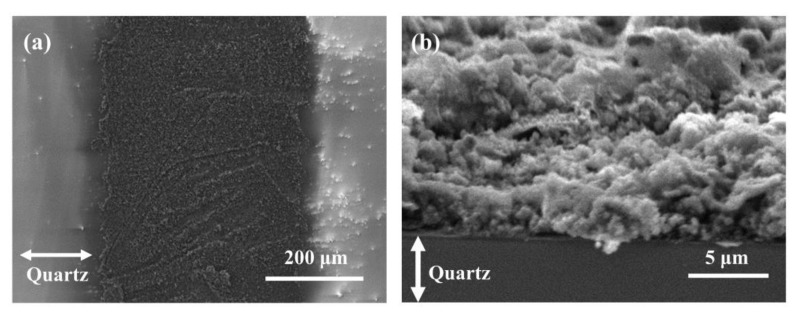
SEM images of Cu@LSC (**a**) in horizontal state and (**b**) tilted at 70°.

**Figure 8 materials-14-03388-f008:**
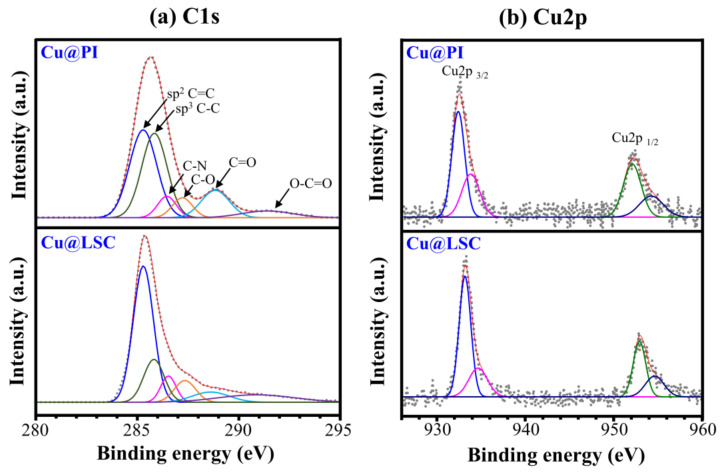
(**a**) C1s and (**b**) Cu2p spectra of Cu@PI and Cu@LSC via XPS analysis.

**Figure 9 materials-14-03388-f009:**
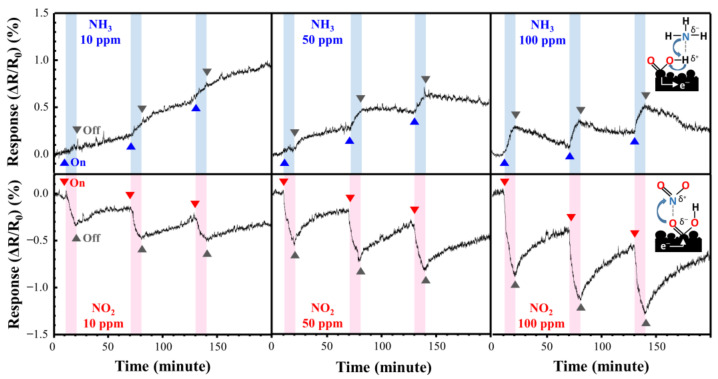
Response curves of Cu@LSC with NO_2_ and NH_3_ exposure.

**Figure 10 materials-14-03388-f010:**
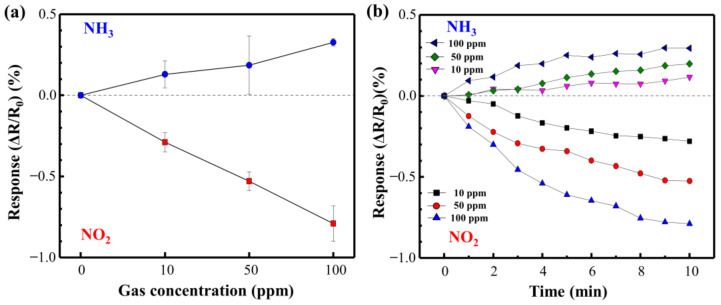
(**a**) Resistance response rates and (**b**) resistance response vs. time for NO_2_ and NH_3_.

## Data Availability

Data sharing not applicable.

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
