# Peer review of "Direct Pattern Growth of Carbon Nanomaterials by Laser Scribing on Spin-Coated Cu-PI Composite Films and Their Gas Sensor Application"

_materials, 2021, doi:10.3390/ma14123388_

Round 1

Reviewer 1 Report

The methodology implemented in the manuscript is well described. The results are represented in detail and discussed, the conclusions are substantiated. Overall, the study is interesting and has practical value. It's worth publishing it in Materials.

As a recommendation, I suggest improving the quality of the figures for a better perception by the readers.

The methodology implemented in the manuscript is well described. The results are represented in detail and discussed, the conclusions are substantiated. Overall, the study is interesting and has practical value. It's worth publishing it in Materials.

There are some comments of mine given below.

  1. As a recommendation, I suggest improving the quality of the figures for a better perception by the readers.
  2. A more detailed explanation of the sensing mechanism of the Cu – laser-scribed carbon composite should be presented.
  3. Have the authors evaluated the selectivity of synthesized Cu@LSC for gases other than NO2 and NH3?

Reviewer 2 Report

The paper “Direct Pattern Growth of Carbon Nanomaterials by Laser Scribing on Spin-Coated Cu-PI Composite Films and its Gas Sensor Application” reported the gas sensing properties of a series of carbon nanomaterials prepared by laser scribing. By applying laser to Cu-embedded PI thin films, gas sensing materials could be produced with low-cost and the production time is greatly reduced. Overall, the material preparation method presented in this work is interesting and has the potential to be scaled up, and the impacts of different processing conditions/parameters on the structure of the obtained carbon nanomaterials were investigated comprehensively. Hence, I recommend the paper to be published after minor revision. Below are my comments that I hope the authors could address properly.

  1. The incorporation ratio of Cu in the Cu-PI thin film has profound impacts on the laser absorbance of the precursor materials and this ratio eventually affects the quality of the obtained carbon nanomaterials. I was wondering if the authors could conduct TGA (in air) to calculate the actual Cu-loading in the Cu-PI thin film as this value might be different than the nominal Cu weight ratio in the casting solution (especially at high Cu loading)?
  2. The obtained carbon nanomaterials exhibited a stereoscopic 3D structure, and such structure is beneficial for gas-sensing applications. Could the authors provide further information on the structure of carbon nanomaterials (e.g., specific surface area, pore volume and/or pore size distribution) using N2 adsorption/desorption tests? Knowing these pieces of information would help identify other gas-sensing targets.
  3. In the gas sensing tests, were the adsorption and desorption of NO2 and NH3 fully reversible under the measurement conditions? Also, how do the gas sensing performances (i.e., detection limit/responding time) of the materials in this work compare to other carbon-based materials (especially the sensors prepared from the classic high temperature method)?
